# Stability of the Maxillary and Mandibular Total Arch Distalization Using Temporary Anchorage Devices (TADs) in Adults

Byung-Jae Song , Kee-Joon Lee, Jung-Yul Cha , Jeong-Seob Lee, Sung-Seo Mo and Hyung-Seog Yu *

Department of Orthodontics, Institute of Craniofacial Deformity, Yonsei University College of Dentistry, Seoul 03722, Korea; coolbjsong@naver.com (B.-J.S.); orthojn@yuhs.ac (K.-J.L.); jungcha@yuhs.ac (J.-Y.C.); orthosub@yonsei.ac.kr (J.-S.L.); dmoss1@hanmail.net (S.-S.M.)
* Correspondence: yumichael@yuhs.ac

**Abstract:** Distalization with temporary anchorage devices (TADs) is commonly used to resolve crowding and to correct molar relationships in non-extraction cases. The purpose of this study was to quantify the treatment effects and post-treatment stability of total arch distalization with TADs in adults and thereby elucidate the clinical effect of this treatment modality. The subjects of the study were 39 adult orthodontic patients treated with total arch distalization with TADs. Lateral cephalograms and dental casts were taken at pretreatment (T0), post-treatment (T1), and the retention period (T2, 29.3 ± 12.8 months) to evaluate the vertical and horizontal movement of teeth, changes of arch width and molar rotation. It was concluded that even though there was a little relapse in the anteroposterior position of the maxillary and mandibular teeth during retention, there was no obvious relapse in the facial profile. Therefore, the total arch distalization can be used in patients with a moderate amount of arch length discrepancy effectively with stable retention.

**Keywords:** total arch distalization; stability; temporary anchorage devices; arch width; molar rotation

## 1. Introduction

Premolar extraction is a method that has been continuously implemented since Tweed proposed to overcome the lack of dentoalveolar discrepancies. However, in patients with moderate crowding, the selection of premolar extraction and non-extraction is made in consideration of the skeletal pattern of the patient and the effect on the facialesthetics. Molar distalization is a nonextraction treatment modality used to correct Class II or Class III molar relationships [1,2] and to relieve crowding without adverse arch expansion and interdental reduction, which can jeopardize both esthetics and stability [3,4]. There have been many attempts to distalize molars with extraoral and intraoral distalizing appliances. The main disadvantages of extraoral anchorage devices such as headgear are the need for patient compliance and the fact that they are esthetically unacceptable [5–7]. To overcome these limitations, many intraoral methods were used to distalize molars. Pendulums [8], distal jets [9], magnets [10], Franzulum appliances [11], and several other methods can be used as intraoral appliances; however, the common and unwanted side effects of intraoral appliances are anchorage loss at the reactive part, flaring of the incisors, distal tipping, and rotation of the distalized molars [12–15].

To reduce the impact of these consequences, the use of temporary anchorage devices (TADs), such as miniplates and miniscrews, has become a new orthodontic treatment strategy over the past decades [16–21]. TADs provide stationary anchorage for various tooth movements without the need for active patient compliance and with no undesirable side effects. The nature of absolute anchorage allows for the retraction of the anterior teeth with simultaneous distal movement of the posterior teeth [22].

Several clinical case reports showed the efficacy of TADs and the efficiency of the treatment mechanics in distalization of the whole dentition. However, there are few studies

with adequate numbers of subjects evaluating the treatment effects of these mechanics with cephalometric analysis, and no study evaluated post-treatment changes of the distalized dentition. Post-treatment stability is not a separate problem in orthodontics but one to be considered in diagnosis and treatment planning [23]. Thus, it is as important to investigate the post-treatment stability of total arch distalization as it is to demonstrate the overall effectiveness of this procedure. Hence, the purpose of this study was to quantify the treatment effects and post-treatment stability of total arch distalization in adults. We also determined whether initial skeletal pattern and treatment changes were correlated with post-treatment changes during retention.

## 2. Materials and Method

### 2.1. Subjects

A sample of thirty-nine adult patients (31 females, 8 males), treated with TADs to distalize dentition at the orthodontic department at Yonsei University Dental Hospital, Seoul, Korea, were selected as subjects in this study. In total, 28 patients had maxillary TADs to distalize the whole maxillary dentition. A total of 25 patients had mandibular TADs, whereas 14 of these 25 patients had maxillary TADs at the same time (Table 1). All patients met the following inclusion criteria: (1) patients older than 18 years at initial status, (2) intact permanent dentition including second molars, (3) without extraction of the premolars or other teeth except the third molars, (4) minimal crowding (<4 mm), (5) followed at least 1 year for post-retention, and (6) no syndrome or skeletal disharmony. The mean age at the beginning of treatment was 25.5 years (range: 18.3–32.3), and the mean treatment period was 24.5 months (range: 16–34 months). The mean period during which total arch distalization force was applied was 12.1 months (range: 6–22 months), and the mean retention period was 29.3 months (range: 14–52 months). All patients were given lingual fixed retainers between the canines and removable circumferential retainers for retention. The Yonsei Dental Hospital institutional review board (CRNo: 2-2020-0013) approved this study, and informed consent agreements were signed by the participants. The minimum sample size was calculated using G*Power 3 (Düsseldorf, Germany) with a significance level of $p < 0.05$ and a power of 80%, and it was confirmed as 25. The descriptive data of the patients are given in Table 1 and the characteristics of the patients are shown in Table 2.

**Table 1.** Descriptive distribution of the patients.

| Patients | Sex | Age | Location of TADs Placement | Duration of Force Application | Duration of Retention |
|---|---|---|---|---|---|
| 1 | F | 25 Y 2 M | #15-16 B, #25-26 B<br>Lt. R, Rt. R | 15 M | 30 M |
| 2 | F | 18 Y 5 M | #16-17 B, #25-26 B<br>#35-36 B, #45-46 B | 9 M | 21 M |
| 3 | F | 19 Y 10 M | #15-16-17 B, #25-26-27 B<br>#36-37 B, #46-47 B | 14 M | 26 M |
| 4 | F | 32 Y 3 M | #16-17 P, #26-27 P<br>#35-36 B, #45-46 B | 18 M | 19 M |
| 5 | F | 25 Y 6 M | #15-16 B, #25-26 B<br>#36-37 B, #46-47 B | 11 M | 48 M |
| 6 | M | 28 Y 6 M | #15-16-17 B, #25-26-27 B<br>#35-36 B, #45-46 B | 22 M | 43 M |
| 7 | F | 25 Y 2 M | #16-17 B, #26-27 B<br>#35-36 B, #45-46 B | 16 M | 22 M |
| 8 | F | 24 Y 11 M | #15-16-17 B, #25-26-27 B<br>#36-37 B, #46-47 B | 12 M | 24 M |
| 9 | M | 24 Y 9 M | #15-16 B, #25-26 B<br>Lt. R, Rt. R | 8 M | 24 M |

**Table 1.** *Cont.*

| Patients | Sex | Age | Location of TADs Placement | Duration of Force Application | Duration of Retention |
|---|---|---|---|---|---|
| 10 | F | 28 Y 4 M | #15-16 B, #25-26 B #36-37 B, #46-47 B | 17 M | 36 M |
| 11 | F | 18 Y 7 M | #15-16 B, #25-26 B #35-36 B, #45-46 B | 13 M | 27 M |
| 12 | F | 28 Y 3 M | #15-16 P, #25-26 P #35-36 B, #45-46 B | 12 M | 24 M |
| 13 | F | 29 Y 7 M | Midpalatal #36-37 B, #46-47 B | 9 M | 18 M |
| 14 | F | 25 Y 9 M | #15-16-17 B, #25-26-27 B #35-36 B, #45-46 B | 21 M | 23 M |
| 15 | M | 20 Y 11 M | #15-16-17 B, #25-26-27 B | 13 M | 47 M |
| 16 | F | 19 Y 10 M | #15-16-17 B, #25-26-27 B | 9 M | 52 M |
| 17 | F | 22 Y 7M | #16-17 P, #26-27 P | 6 M | 50 M |
| 18 | F | 18 Y 7M | #15-16 B, #25-26 B | 18 M | 31 M |
| 19 | F | 20 Y 6M | #15-16 B, #25-26 B | 13 M | 24 M |
| 20 | M | 18 Y 9M | #15-16 B, #25-26 B | 14 M | 13 M |
| 21 | F | 20 Y 2 M | #15-16-17 B, #25-26-27 B | 12 M | 23 M |
| 22 | F | 32 Y 3 M | #16-17 B, #26-27 B | 14 M | 20 M |
| 23 | F | 35 Y 3 M | #15-16 B, #25-26 B | 9 M | 25 M |
| 24 | F | 23Y 11 M | #15-16 B, #25-26 B | 8 M | 37 M |
| 25 | F | 31 Y 8 M | #15-16-17 B, #25-26-27 B | 7 M | 25 M |
| 26 | M | 32 Y 2 M | #15-16 B, #25-26 B | 22 M | 41 M |
| 27 | F | 23 Y 8 M | #15-16 B, #25-26 B | 7 M | 14 M |
| 28 | F | 31 Y 8 M | #16-17 B, #26-27 B | 7 M | 25 M |
| 29 | F | 18 Y 3 M | #35-36 B, #45-46 B | 20 M | 28 M |
| 30 | M | 18 Y 9 M | #35-36 B, #45-46 B | 9 M | 29 M |
| 31 | F | 24 Y 6 M | #36-37 B, #46-47 B | 8 M | 33 M |
| 32 | F | 22 Y 9 M | Lt. R, Rt. R | 12 M | 24 M |
| 33 | F | 22 Y 1 M | Lt. R, Rt. R | 12 M | 17 M |
| 34 | F | 24 Y 3 M | #35-36 B, #45-46 B | 18 M | 38 M |
| 35 | M | 29 Y 7 M | #36-37 B, #46-47 B | 13 M | 41 M |
| 36 | F | 22 Y 11 M | Lt. R, Rt. R | 11 M | 20 M |
| 37 | F | 18 Y 3 M | #35-36 B, #45-46 B | 20 M | 28 M |
| 38 | M | 18 Y 9 M | #35-36 B, #45-46 B | 9 M | 29 M |
| 39 | F | 22 Y 9 M | #36-37 B, #46-47 B | 19 M | 39 M |

M: male, F: female, Y: years, M: months, B: buccal, P: palatal, R: ramus.

**Table 2.** Characteristics of patients.

| Variables | |
|---|---|
| Sex | 8 male (20.5)/31 female (79.5) |
| Age (years) (mean ± SD) | 24.5 ± 5.38 |
| Crowding (mm) (mean ± SD) | |
|     Maxilla | 2.43 ± 0.89 |
|     Mandible | 1.81 ± 0.51 |
| Sagittal skeletal pattern | |
|     Class I | 3 (8.0) |
|     Class II | 19 (48.5) |
|     Class III | 17 (43.5) |
| Vertical skeletal pattern | |
|     Normal (SN-MP 27–37°) | 22 (56.5) |
|     High mandibular plane angle (>37°) | 11 (28.0) |
|     Low mandibular plane angle (<27°) | 6 (15.5) |

**Table 2.** *Cont.*

| Variables | |
|---|---|
| Distalization arch | |
|     Maxillary arch only | 14 (36.0) |
|     Mandibular arch only | 11 (28.0) |
|     Both maxillary and mandibular arch | 14 (36.0) |
| A number of TADs by insertion sites | |
|     Between 2nd premolar and 1st molar | 60 (54.5) |
|     Between 1st molar and 2nd molar | 32 (29.0) |
|     Ramus | 10 (9.0) |
|     Midpalate | 2 (1.5) |

SN-MP, mandibular plane angle. Unless otherwise noted, the right column means number (%).

## 2.2. Appliances and TADs

Pre-adjusted 0.018 × 0.025-inch slot edgewise appliances with Roth prescription (Tomy, Tokyo, Japan) was used in all patients and approximately 200 cN of distalizing forces were applied by ligating nickel titanium closed-coil springs or elastic chains (Ormco, Glendora, CA, USA) from the maxillary and mandibular TADs (Ortholution, Seoul, Korea) to the canines or premolars in the maxillary and mandibular arches. Posterior teeth were distalized first if there was moderate crowding to resolve before whole arch distalization (Figure 1a). During distalization, the main archwire was 0.016 × 0.022-inch stainless steel in the maxilla and the mandible (Figure 1b). In very few cases, screw placement failed. In case the screw fails, re-implantation proceeds as soon as possible so that the entire treatment period is not affected.

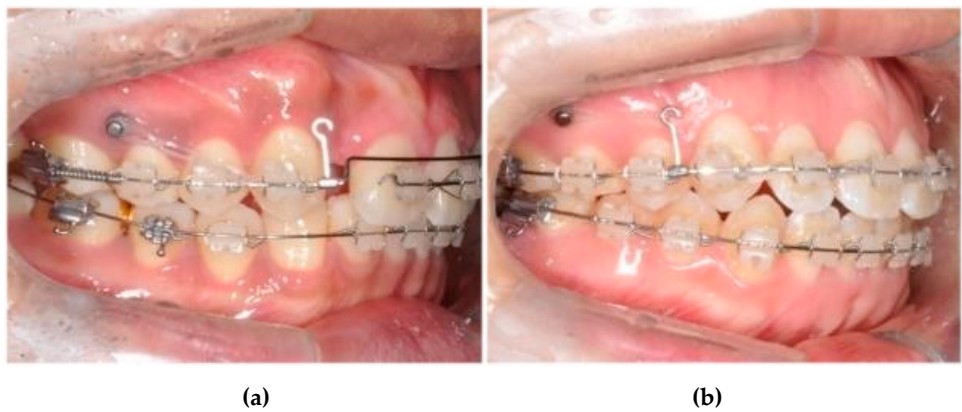

|       (a)       |       (b)       |

**Figure 1.** Partial canine retraction (**a**) and en-masse retraction (**b**).

In the maxilla, 48 miniscrews were inserted into the buccal alveolar bone between the second premolar and the first molars, and 28 miniscrews were inserted into the buccal alveolar bone between the first molar and second molar. Two miniscrews were placed in the midpalatal area. In the mandible, 16 miniscrews and 6 miniplates were inserted into the buccal alveolar bone between the mandibular first molar and second molar, 18 into the buccal alveolar bone between the mandibular second premolar and the first molar, and 10 into the retromolar area. Distalization force was stopped when desired occlusion and facial profile was obtained.

## 2.3. Cephalometric Measurements

Pretreatment (T0), post-treatment (T1) and post-retention (T2) cephalograms were taken with the Cranex 3+ (Soredex, Helsinki, Finland), and digitized using V-ceph program (Osstem Inc., Seoul, Korea). All lateral cephalograms were traced by one examiner and the intra-individual method error did not exceed 0.2 mm. In total, 11 angular and 29 linear measurements were calculated to evaluate skeletal, dental and soft tissue changes before

distalization, after distalization and during the retention period. When there was a double image, the midpoint between the 2 superimposed points was selected. The pterygoid vertical (PTV) plane was used to determine the amount of horizontal movement of maxillary and mandibular teeth [24]. For the vertical movement of the maxillary and mandibular teeth, superimposition of the palatal plane (PP) and mandibular plane (MP) was used, respectively. Angular changes of tooth positions were determined by the inclination of the teeth to the sella-nasion plane (SN) in the maxilla and to the MP in the mandible. The skeletal and soft tissue measurements, dental linear measurement, and dental angular measurements are illustrated in Figures 2–5.

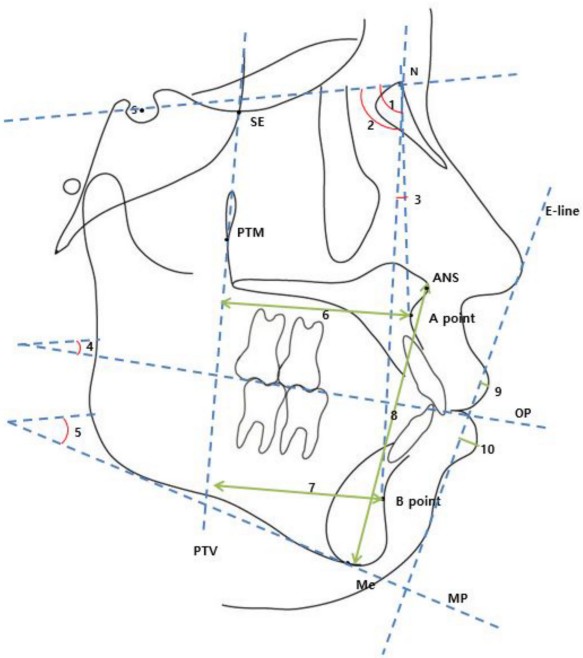

**Figure 2.** Cephalometric skeletal and soft tissue measurements. 1, SNA; 2, SNB; 3, ANB; 4, SN-OP (occlusal plane angle); 5, SN-MP (mandibular plane angle); 6, PTV to A point; 7, PTV to B point; 8, ANS-Me (lower anterior facial height); 9, upper lip to E-line; 10, lower lip to E-line.

*2.4. Model Measurements*

Dental changes of the distalized maxilla and mandibular arches were measured before treatment, after treatment and during the post-retention period with dental casts using Geomagic Control (3D systems, Rock Hill, SC, USA). Intercanine width and intermolar width were measured to evaluate arch expansion. To evaluate the rotation of the molars, an angle between perpendicular line to the central groove of left and right molars was measured (Figure 6).

*2.5. Statistical Analysis*

All statistical analyses were performed using IBM SPSS Statistics software for Windows, version 20.0 (SPSS Inc., Chicago, IL, USA). With a 2-week interval, all cephalometric digitizing and analyses were repeated by the same examiner. The method error was calculated by using the intraclass correlation coefficient (ICC), which was ranged between 0.963 and 0.915 for all cephalometric and cast variables measured in this study.

The Kolmogorov–Smirnov test was used to confirm the normality of the data distribution. The repeated measures analysis of variance (RMANOVA) was then used to determine the treatment and post-treatment changes over time (T0, T1 and T2). Additionally, Pearson's correlation coefficients were calculated to verify the association between post-treatment dental changes and other variables.

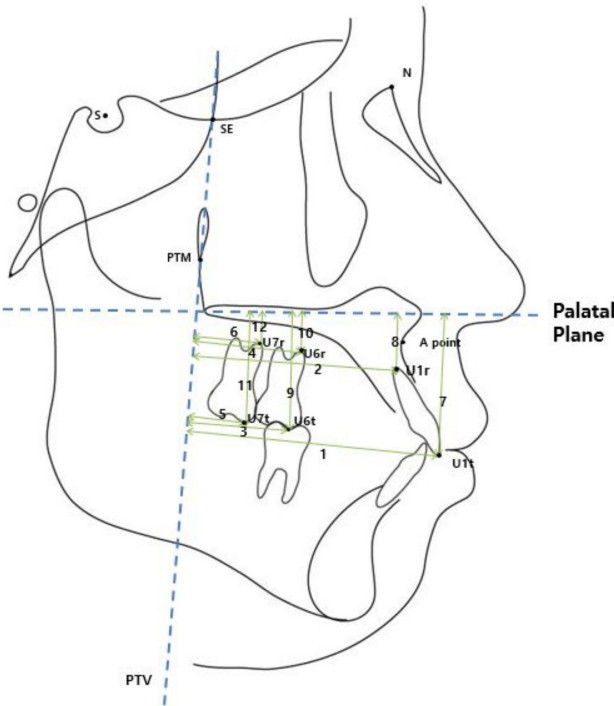

**Figure 3.** Cephalometric dental linear measurements of maxilla. Horizontal measurements: 1, PTV to incisor tip; 2, PTV to incisor root; 3, PTV to first molar cusp; 4, PTV to first molar root; 5, PTV to second molar cusp; 6, PTV to second molar root. Vertical measurements: 7, PP (palatal plane) to incisor tip; 8, PP to incisor root; 9, PP to first molar cusp; 10, PP to first molar root; 11, PP to second molar cusp; 12, PP to second molar root.

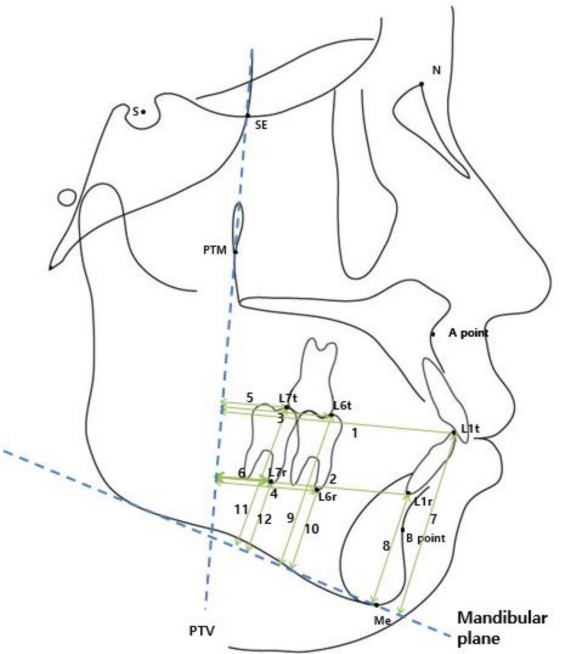

**Figure 4.** Cephalometric dental linear measurements of mandible. Horizontal measurements: 1, PTV to incisor tip; 2, PTV to incisor root; 3, PTV to first molar cusp; 4, PTV to first molar root; 5, PTV to second molar cusp; 6, PTV to second molar root. Vertical measurements: 7, MP (mandibular plane) to incisor tip; 8, MP to incisor root; 9, MP to first molar cusp; 10, MP to first molar root; 11, MP to second molar cusp; 12, MP to second molar root.

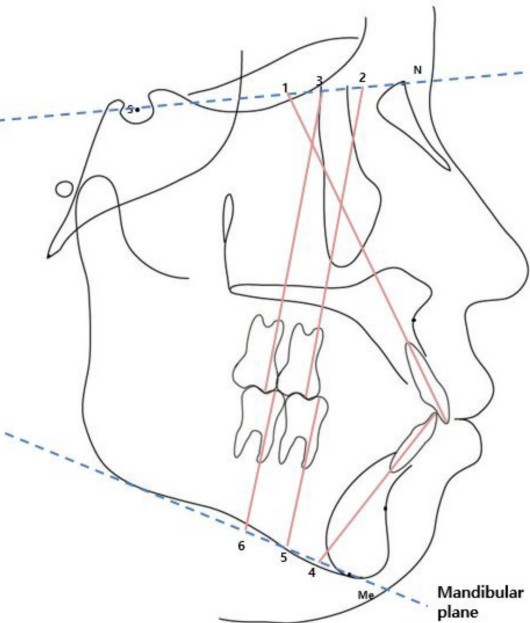

**Figure 5.** Cephalometric dental angular measurements. In maxilla: 1, SN to upper incisor; 2, SN to upper first molar; 3, SN to upper second molar. In mandible: 4, MP to lower incisor; 5, MP to lower first molar; 6, MP to lower second molar.

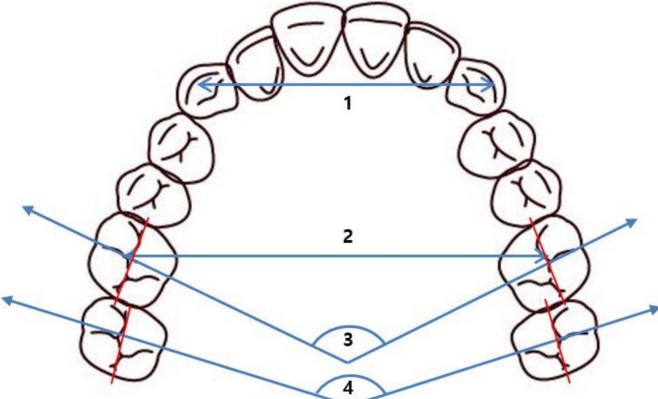

**Figure 6.** Model measurements. 1, ICW (intercanine width); 2, IMW (intermolar width); 3, Rotation of first molar; 4, Rotation of second molar.

## 3. Result

### 3.1. Skeletal Changes

The skeletal changes during and after distalization are summarized in Tables 3 and 4. For the result of maxillary arch distalization, lower anterior facial height (ANS-Me) decreased significantly for 0.74 mm and kept relatively stable during the retention period. The other measurements were not statistically different.

**Table 3.** Descriptive statistics of cephalometric measurements at pretreatment, post-treatment, postretention, pretreatment to post-treatment (T1-T0), post-treatment to postretention (T2-T1) and pretreatment to postretention (T2-T0) of maxillary arch distalization group.

| | T0 | | T1 | | T2 | | T1-T0 | | | T2-T1 | | | T2-T0 | | |
|---|---|---|---|---|---|---|---|---|---|---|---|---|---|---|---|
| | Mean | SD | Mean | SD | Mean | SD | Mean | SD | Sig | Mean | SD | Sig | Mean | SD | Sig |
| **Skeletal** | | | | | | | | | | | | | | | |
| **SNA(°)** | 80.89 | 3.18 | 80.60 | 2.80 | 80.66 | 2.93 | −0.29 | 1.65 | | 0.06 | 1.46 | | −0.23 | 1.26 | |
| **SNB(°)** | 75.82 | 3.55 | 75.68 | 3.37 | 75.83 | 3.57 | −0.13 | 1.38 | | 0.15 | 1.38 | | 0.01 | 0.92 | |
| **ANB(°)** | 5.07 | 2.04 | 4.91 | 1.94 | 4.82 | 1.90 | −0.16 | 0.95 | | −0.09 | 0.54 | | −0.25 | 0.98 | |
| **SN-OP(°)** | 21.70 | 4.49 | 22.77 | 4.73 | 22.35 | 4.81 | 1.06 | 2.24 | | −0.41 | 1.62 | | 0.65 | 2.30 | |
| **SN-MP(°)** | 39.21 | 6.32 | 39.21 | 6.34 | 39.44 | 6.46 | 0.01 | 1.61 | | 0.23 | 2.03 | | 0.23 | 2.13 | |
| **PTV-A(mm)** | 52.59 | 3.22 | 52.42 | 3.00 | 52.28 | 3.16 | −0.18 | 1.79 | | −0.14 | 1.76 | | −0.31 | 2.24 | |
| **PTV-B(mm)** | 50.42 | 6.57 | 50.23 | 6.30 | 49.95 | 7.05 | −0.19 | 2.36 | | −0.28 | 2.55 | | −0.47 | 2.39 | |
| **ANS-Me(mm)** | 75.86 | 5.09 | 75.12 | 5.43 | 75.44 | 5.53 | −0.74 | 1.47 | * | 0.32 | 1.77 | | −0.42 | 2.41 | * |
| **Facial Height Ratio(%)** | 62.65 | 4.74 | 62.67 | 4.87 | 62.34 | 5.16 | 0.01 | 0.98 | | −0.32 | 2.02 | | −0.31 | 2.08 | |
| **Dental linear (mm)** | | | | | | | | | | | | | | | |
| **U1t-PTV** | 63.13 | 4.17 | 60.45 | 4.44 | 60.88 | 4.79 | −2.68 | 2.19 | *** | 0.43 | 1.15 | | −2.25 | 2.46 | *** |
| **U1r-PTV** | 49.61 | 4.31 | 48.53 | 3.84 | 48.38 | 3.88 | −1.09 | 2.11 | * | −0.15 | 1.33 | | −1.23 | 3.07 | * |
| **U6t-PTV** | 28.39 | 4.16 | 25.93 | 4.11 | 26.45 | 3.87 | −2.46 | 1.97 | *** | 0.52 | 0.99 | * | −1.94 | 1.72 | *** |
| **U6r-PTV** | 27.98 | 2.84 | 26.35 | 3.33 | 26.80 | 2.78 | −1.63 | 2.14 | ** | 0.45 | 0.79 | | −1.18 | 1.75 | ** |
| **U7t-PTV** | 17.55 | 4.27 | 14.95 | 4.34 | 15.60 | 3.91 | −2.60 | 2.52 | *** | 0.65 | 0.92 | * | −1.95 | 2.05 | ** |
| **U7r-PTV** | 18.26 | 2.78 | 15.82 | 2.89 | 16.40 | 2.51 | −2.44 | 2.34 | *** | 0.58 | 1.03 | * | −1.86 | 2.09 | *** |
| **U1t-PP** | 33.03 | 3.18 | 32.74 | 3.04 | 33.00 | 13.07 | −0.28 | 1.40 | | 0.26 | 2.57 | | −0.03 | 1.97 | |
| **U1r-PP** | 11.52 | 3.43 | 10.91 | 2.76 | 11.07 | 2.99 | −0.61 | 2.45 | | 0.16 | 0.76 | | −0.45 | 1.24 | |
| **U6t-PP** | 25.03 | 2.02 | 24.11 | 2.35 | 24.46 | 2.54 | −0.92 | 1.16 | ** | 0.35 | 0.55 | | −0.57 | 1.51 | ** |
| **U6r-PP** | 7.81 | 1.62 | 6.45 | 2.02 | 6.86 | 2.29 | −1.36 | 1.52 | *** | 0.41 | 0.47 | | −0.95 | 1.76 | ** |
| **U7t-PP** | 22.14 | 2.22 | 21.25 | 2.54 | 21.63 | 2.87 | −0.89 | 1.17 | ** | 0.38 | 0.89 | | −0.51 | 1.76 | * |
| **U7r-PP** | 5.39 | 1.69 | 4.04 | 1.93 | 4.48 | 2.08 | −1.35 | 1.39 | *** | 0.44 | 0.62 | | −0.91 | 1.49 | * |
| **Dental angular(°)** | | | | | | | | | | | | | | | |
| **U1 to SN** | 105.56 | 6.24 | 101.70 | 6.91 | 102.79 | 7.57 | −3.86 | 4.28 | *** | 1.08 | 3.55 | | −2.77 | 4.11 | ** |
| **U6 to SN** | 72.59 | 5.72 | 69.94 | 5.74 | 71.55 | 6.05 | −2.66 | 3.97 | ** | 0.71 | 2.44 | * | −1.04 | 3.68 | * |
| **U7 to SN** | 68.81 | 7.58 | 68.32 | 6.16 | 68.78 | 6.66 | −0.50 | 5.42 | | 0.46 | 2.02 | * | −0.04 | 5.68 | |
| **Soft tissue (mm)** | | | | | | | | | | | | | | | |
| **Upper Lip E-plane** | 0.80 | 1.85 | −0.10 | 1.85 | 0.06 | 1.58 | −0.89 | 1.19 | *** | 0.16 | 0.89 | | −0.73 | 1.12 | *** |
| **Lower Lip E-plane** | 2.17 | 2.18 | 1.32 | 2.07 | 1.28 | 1.92 | −0.85 | 2.08 | * | 0.04 | 1.07 | | −0.81 | 1.82 | * |

T0, pretreatment; T1, post-treatment; T2, postretention; SD, standard deviation; Sig, significance. * $p < 0.05$; ** $p < 0.01$; *** $p < 0.001$.

**Table 4.** Descriptive statistics of cephalometric measurements at pretreatment, post-treatment, postretention, pretreatment to post-treatment (T1-T0), post-treatment to postretention (T2-T1) and pretreatment to postretention (T2-T0) of mandibular arch distalization group.

| | T0 | | T1 | | T2 | | T1-T0 | | | T2-T1 | | | T2-T0 | | |
|---|---|---|---|---|---|---|---|---|---|---|---|---|---|---|---|
| | Mean | SD | Mean | SD | Mean | SD | Mean | SD | Sig | Mean | SD | Sig | Mean | SD | Sig |
| **Skeletal** | | | | | | | | | | | | | | | |
| **SNA(°)** | 80.20 | 2.88 | 80.47 | 2.80 | 80.32 | 3.26 | 0.27 | 0.68 | | −0.14 | 1.30 | | 0.12 | 1.64 | |
| **SNB(°)** | 76.81 | 3.48 | 76.91 | 3.67 | 76.89 | 3.53 | 0.09 | 0.63 | | −0.02 | 1.12 | | 0.08 | 1.28 | |
| **ANB(°)** | 3.39 | 2.67 | 3.56 | 2.68 | 3.43 | 2.63 | 0.17 | 0.64 | | −0.13 | 0.59 | | 0.04 | 0.82 | |
| **SN-OP(°)** | 20.94 | 4.88 | 21.38 | 5.82 | 21.07 | 5.36 | 0.44 | 2.56 | | −0.31 | 1.66 | | 0.13 | 2.99 | |
| **SN-MP(°)** | 37.73 | 6.61 | 37.34 | 6.48 | 37.87 | 6.56 | −0.38 | 0.91 | | 0.53 | 2.01 | | 0.14 | 2.34 | |
| **PTV-A(mm)** | 51.96 | 2.31 | 52.78 | 2.71 | 52.69 | 3.45 | 0.82 | 2.45 | | −0.08 | 1.70 | | 0.73 | 2.76 | |
| **PTV-B(mm)** | 52.50 | 6.98 | 52.15 | 7.08 | 52.03 | 7.52 | −0.35 | 2.03 | * | −0.12 | 2.34 | | −0.47 | 3.73 | * |
| **ANS-Me(mm)** | 74.63 | 6.65 | 74.57 | 5.80 | 75.52 | 5.81 | −0.06 | 2.00 | | 0.95 | 1.91 | | 0.89 | 2.75 | |
| **Facial Height Ratio(%)** | 63.26 | 5.17 | 63.52 | 5.21 | 63.09 | 5.42 | 0.26 | 0.93 | | −0.43 | 2.19 | | −0.17 | 2.31 | |

**Table 4.** *Cont.*

| | T0 | | T1 | | T2 | | T1-T0 | | | T2-T1 | | | T2-T0 | | |
|---|---|---|---|---|---|---|---|---|---|---|---|---|---|---|---|
| | Mean | SD | Mean | SD | Mean | SD | Mean | SD | Sig | Mean | SD | Sig | Mean | SD | Sig |
| **Dental linear (mm)** | | | | | | | | | | | | | | | |
| L1t-PTV | 58.76 | 4.10 | 57.85 | 4.03 | 58.39 | 4.73 | −0.91 | 3.43 | * | 0.53 | 0.94 | | −0.38 | 3.60 | |
| L1r-PTV | 50.11 | 6.02 | 49.63 | 6.23 | 49.95 | 7.06 | −0.48 | 4.45 | | 0.32 | 1.95 | | −0.16 | 4.76 | |
| L6t-PTV | 30.71 | 3.93 | 28.14 | 3.90 | 28.72 | 3.90 | −2.57 | 4.13 | ** | 0.58 | 1.06 | | −1.99 | 3.16 | ** |
| L6r-PTV | 26.88 | 6.01 | 26.01 | 5.66 | 26.23 | 5.72 | −0.88 | 4.23 | | 0.22 | 1.26 | | −0.65 | 3.56 | |
| L7t-PTV | 18.72 | 4.05 | 16.48 | 3.93 | 16.90 | 3.81 | −2.24 | 4.35 | * | 0.42 | 0.90 | | −1.82 | 3.37 | * |
| L7r-PTV | 13.80 | 5.94 | 13.72 | 5.47 | 13.94 | 5.29 | −0.09 | 4.45 | | 0.23 | 1.16 | | 0.14 | 3.62 | |
| L1t-MP | 45.49 | 4.78 | 44.36 | 4.06 | 44.91 | 4.06 | −1.13 | 2.63 | | 0.55 | 0.63 | | −0.58 | 2.99 | |
| L1r-MP | 25.29 | 4.52 | 24.77 | 3.25 | 25.33 | 3.68 | −0.52 | 2.37 | | 0.57 | 1.08 | | 0.04 | 3.43 | |
| L6t-MP | 35.48 | 3.27 | 35.22 | 3.04 | 35.25 | 3.29 | −0.26 | 1.65 | | 0.04 | 0.63 | | −0.23 | 1.98 | |
| L6r-MP | 16.28 | 2.72 | 16.95 | 3.10 | 17.12 | 3.09 | 0.67 | 2.12 | | 0.17 | 0.58 | | 0.84 | 1.72 | |
| L7t-MP | 32.62 | 2.83 | 32.18 | 2.96 | 32.00 | 3.47 | −0.44 | 1.51 | | −0.18 | 0.74 | | −0.62 | 2.08 | |
| L7r-MP | 15.24 | 2.68 | 14.95 | 2.86 | 14.93 | 2.98 | −0.30 | 1.82 | | −0.01 | 0.52 | | −0.31 | 1.54 | |
| **Dental angular(°)** | | | | | | | | | | | | | | | |
| L1 to MP | 97.07 | 5.65 | 95.39 | 5.74 | 96.05 | 4.74 | −1.68 | 5.01 | | 0.66 | 3.36 | | −1.02 | 4.75 | |
| L6 to MP | 75.39 | 7.36 | 73.09 | 7.77 | 73.48 | 7.83 | −2.30 | 6.05 | | 0.39 | 4.15 | | −1.91 | 6.90 | |
| L7 to MP | 87.44 | 4.59 | 80.22 | 3.81 | 81.06 | 3.46 | −7.21 | 4.99 | *** | 0.84 | 3.53 | | −6.38 | 3.87 | *** |
| **Soft tissue (mm)** | | | | | | | | | | | | | | | |
| Upper Lip E-plane | −0.25 | 2.11 | −0.88 | 2.26 | −0.63 | 2.26 | −0.63 | 1.24 | * | 0.25 | 0.88 | | −0.38 | 1.23 | * |
| Lower Lip E-plane | 1.67 | 2.58 | 0.62 | 2.62 | 0.85 | 2.44 | −1.06 | 1.91 | * | 0.23 | 1.21 | | −0.83 | 1.42 | * |

T0, pretreatment; T1, post-treatment; T2, postretention; SD, standard deviation; Sig, significance. * $p < 0.05$; ** $p < 0.01$; *** $p < 0.001$.

For the result of mandibular arch distalization, the distance from PTV to B point (PTV-B) decreased by 0.35 mm during treatment and, slightly, decreased again during the retention period.

### 3.2. Dental Changes

In the evaluation of dental variables, in relation to the PTV, PP and MP as the reference lines, there was a significant 2.46 mm distal movement of the maxillary first molars, while their roots were distalized by 1.63 mm with a distal tipping of 2.66°. The incisors showed a significant retraction of 2.68 mm with a lingual inclination of 3.86°. The first and second molars showed significant intrusion values of 0.92 and 0.89 mm, respectively, in the vertical position but not the incisors. After the retention period, maxillary first and second molars moved mesially 0.52 and 0.65 mm with extrusion values of 0.35 and 0.38 mm and labial tipping values of 0.71° and 0.46°, respectively, but the amount was relatively small.

In the mandible, there was a significant 2.57 mm distalization of the first molars, while their roots were distalized by 0.88 mm with a distal tipping of 2.30°. Especially in the mandible, the second molars showed a significantly large amount of distal tipping—7.21°. The incisors showed a significant retraction of 0.91 mm and lingual inclination was not statistically significant. The incisors and molars in the mandible showed no significant change in the vertical position. After the retention period, the entire arch moved mesially, but none of the variables showed statistically significant results. Changes after treatment and retention period are shown in Figures 7 and 8 so that clinicians can compare briefly.

### 3.3. Soft Tissue Changes

The upper and lower lips relative to the E-line showed significant retraction values of 0.89 and 1.06 mm, respectively. The lower lip moved distally more than the upper lip. There was no significant upper and lower lip position change during the retention period.

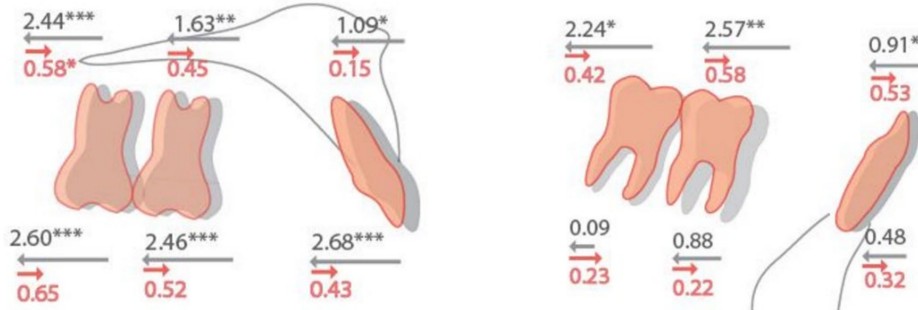

**Figure 7.** The schema about comparison of pretreatment to post-treatment (T1-T0) and post-treatment to postretention (T2-T1). * *p* < 0.05; ** *p* < 0.01; *** *p* < 0.001.

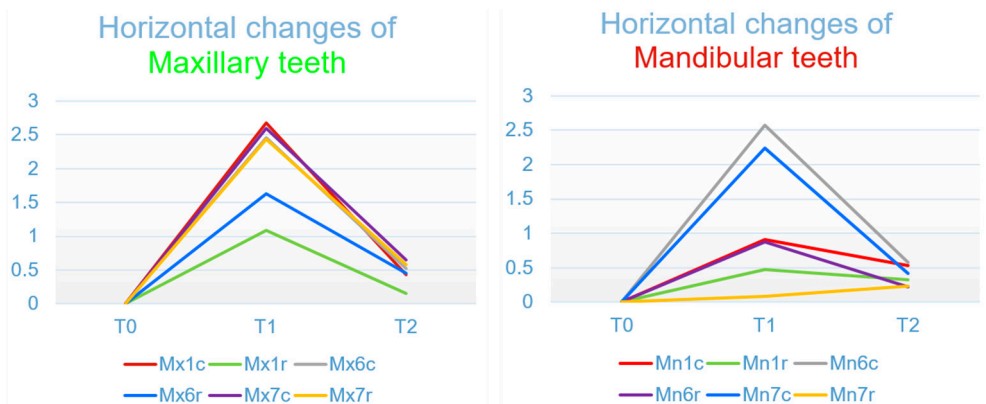

**Figure 8.** Horizontal changes of maxillary and mandibular teeth.

### 3.4. Changes of Arch Width and Molar Rotation

The changes of arch width and molar rotation as measured from dental casts are shown in Table 5. There was a significant difference in maxillary arch width before and after distalization. There were expansions of 1.52 and 0.93 mm in the maxillary canine and first molar, respectively, and 0.88 and 1.14 mm in the mandibular canine and first molar, respectively. Both intercanine width and intermolar width kept relatively stable during the retention period. Distal-in rotation of the first and second molars was observed after the distalization, but it was not statistically significant and it did not change during the retention period either.

**Table 5.** Model measurements at pretreatment, post-treatment, postretention, pretreatment to post-treatment (T1-T0), post-treatment to postretention (T2-T1) and pretreatment to postretention (T2-T0).

| | T0 | | T1 | | T2 | | T1-T0 | | | T2-T1 | | | T2-T0 | | |
| --- | --- | --- | --- | --- | --- | --- | --- | --- | --- | --- | --- | --- | --- | --- | --- |
| | **Mean** | **SD** | **Mean** | **SD** | **Mean** | **SD** | **Mean** | **SD** | **Sig** | **Mean** | **SD** | **Sig** | **Mean** | **SD** | **Sig** |
| **Inter-canine width(mm)** | | | | | | | | | | | | | | | |
| Mx. | 35.26 | 3.53 | 36.78 | 2.56 | 36.13 | 3.06 | 1.52 | 0.33 | *** | −0.65 | 0.71 | | 0.87 | 0.75 | ** |
| Mn. | 27.07 | 3.7 | 27.95 | 2.74 | 27.65 | 2.24 | 0.88 | 1.14 | | −0.35 | 0.55 | | 0.58 | 0.98 | |
| **Inter-molar width(mm)** | | | | | | | | | | | | | | | |
| Mx. | 45.36 | 2.34 | 46.29 | 2.51 | 45.99 | 2.68 | 0.93 | 0.31 | ** | −0.30 | 1.21 | | 0.63 | 1.30 | * |
| Mn. | 44.24 | 3.67 | 45.38 | 3.36 | 45.06 | 3.47 | 1.14 | 1.05 | | −0.32 | 2.95 | | 0.82 | 1.75 | |
| **Molar rotation (°)** | | | | | | | | | | | | | | | |
| **U6rot** | 138.85 | 11.64 | 143.72 | 9.28 | 140.57 | 11.38 | 4.87 | 2.36 | | −3.15 | 1.36 | | 1.72 | 1.62 | |
| **U7rot** | 148.82 | 13.74 | 156.32 | 16.12 | 154.69 | 9.46 | 7.50 | 4.10 | | −1.63 | 2.12 | | 5.87 | 2.21 | |
| **L6rot** | 156.28 | 9.36 | 159.32 | 15.32 | 157.36 | 17.33 | 3.04 | 2.32 | | −1.96 | 2.36 | | 1.08 | 1.71 | |
| **L7rot** | 161.34 | 7.35 | 166.35 | 12.25 | 163.15 | 11.95 | 5.01 | 2.95 | | −4.20 | 1.59 | | 1.81 | 0.96 | |

Mx, maxilla; Mn, mandible; rot, rotation; SD, standard deviation; Sig, significance. * *p* < 0.05; ** *p* < 0.01; *** *p* < 0.001.

### 3.5. Pearson's Correlation Coefficients between Post-Treatment Changes and Other Variables (Initial Skeletal Pattern, Retention Period and Treatment Changes)

In the maxilla, post-treatment horizontal relapse of the first and second molar was significantly negatively correlated with initial ANB and amounts of distalization—that is, the larger the distal movement during the treatment period, the more the mesial drift during the retention period (Table 6).

**Table 6.** Pearson's correlation coefficients between post-treatment changes and other variables in the maxilla.

| | T2-T1 | | | | | | | | | |
|---|---|---|---|---|---|---|---|---|---|---|
| | ΔU1t-PTV | | ΔU6t-PTV | | ΔU7t-PTV | | ΔU6-SN | | ΔU7-SN | |
| | r | Sig | r | Sig | r | Sig | r | Sig | r | Sig |
| **Skeletal pattern (T0)** | | | | | | | | | | |
| ANB | **−0.419 *** | 0.030 | **−0.437 *** | 0.023 | **−0.401 *** | 0.038 | 0.074 | 0.714 | −0.068 | 0.735 |
| SN-MP | −0.109 | 0.587 | 0.011 | 0.956 | 0.008 | 0.968 | −0.047 | 0.815 | −0.174 | 0.386 |
| Facial Height Ratio | 0.029 | 0.884 | −0.105 | 0.602 | −0.077 | 0.704 | 0.008 | 0.967 | 0.164 | 0.413 |
| Retention Period | 0.050 | 0.806 | 0.093 | 0.643 | 0.144 | 0.472 | 0.286 | 0.149 | 0.166 | 0.408 |
| **Treatment changes (T1-T0)** | | | | | | | | | | |
| ΔU1t-PTV | −0.355 | 0.069 | **−0.417 *** | 0.031 | **−0.498 **** | 0.008 | −0.274 | 0.166 | −0.120 | 0.553 |
| ΔU6t-PTV | **−0.543 **** | 0.003 | **−0.623 **** | 0.001 | **−0.674 ***** | 0.000 | −0.093 | 0.644 | −0.121 | 0.549 |
| ΔU7t-PTV | **−0.399 *** | 0.039 | **−0.474 *** | 0.013 | **−0.599 **** | 0.001 | −0.229 | 0.251 | −0.176 | 0.381 |
| ΔU6-SN | 0.107 | 0.595 | 0.219 | 0.272 | 0.165 | 0.412 | −0.252 | 0.204 | 0.010 | 0.960 |
| ΔU7-SN | −0.187 | 0.350 | −0.016 | 0.936 | −0.138 | 0.492 | −0.143 | 0.476 | −0.191 | 0.341 |
| ΔU6rot | 0.205 | 0.306 | −0.181 | 0.377 | −0.038 | 0.851 | −0.010 | 0.961 | −0.138 | 0.491 |
| ΔU7rot | −0.001 | 0.997 | 0.169 | 0.400 | −0.145 | 0.469 | −0.198 | 0.374 | −0.138 | 0.751 |

$* p < 0.05; ** p < 0.01; *** p < 0.001.$

In the mandible, post-treatment horizontal relapse of the first and second molar was not significantly correlated with initial skeletal pattern and retention period. However, they were significantly negatively correlated with amounts of distalization. In particular, angular relapse of the lower second molar was significantly negatively correlated with the amounts of distal tipping (Table 7).

**Table 7.** Pearson's correlation coefficients between post-treatment changes and other variables in the mandible.

| | T2-T1 | | | | | | | | | |
|---|---|---|---|---|---|---|---|---|---|---|
| | ΔL1t-PTV | | ΔL6t-PTV | | ΔL7t-PTV | | ΔL6-MP | | ΔL7-MP | |
| | r | Sig | r | Sig | r | Sig | r | Sig | r | Sig |
| **Skeletal pattern (T0)** | | | | | | | | | | |
| ANB | −0.027 | 0.906 | −0.122 | 0.587 | −0.079 | 0.727 | 0.121 | 0.593 | 0.043 | 0.850 |
| SN-MP | −0.074 | 0.743 | 0.225 | 0.314 | 0.242 | 0.277 | 0.038 | 0.868 | 0.094 | 0.677 |
| Facial Height Ratio | −0.004 | 0.985 | −0.268 | 0.228 | −0.262 | 0.239 | −0.006 | 0.980 | −0.112 | 0.618 |
| Retention Period | −0.318 | 0.149 | −0.224 | 0.315 | 0.012 | 0.957 | −0.113 | 0.617 | −0.144 | 0.522 |
| **Treatment changes (T1-T0)** | | | | | | | | | | |
| ΔL1t-PTV | −0.188 | 0.402 | **−0.572 **** | 0.005 | **−0.613 **** | 0.002 | 0.170 | 0.450 | −0.102 | 0.650 |
| ΔL6t-PTV | −0.131 | 0.562 | **−0.664 **** | 0.001 | **−0.707 ***** | 0.000 | 0.127 | 0.573 | −0.200 | 0.372 |
| ΔL7t-PTV | −0.067 | 0.766 | **−0.550 **** | 0.008 | **−0.673 **** | 0.001 | 0.067 | 0.765 | −0.251 | 0.259 |
| ΔL6-MP | −0.083 | 0.713 | 0.205 | 0.361 | 0.235 | 0.292 | −0.120 | 0.593 | 0.110 | 0.625 |
| ΔL7-MP | 0.007 | 0.976 | −0.241 | 0.280 | −0.272 | 0.222 | −0.121 | 0.590 | **−0.634 **** | 0.002 |
| ΔL6rot | 0.002 | 0.993 | 0.103 | 0.649 | 0.093 | 0.682 | −0.068 | 0.765 | −0.283 | 0.202 |
| ΔL7rot | −0.209 | 0.351 | −0.186 | 0.408 | −0.120 | 0.595 | −0.429 | 0.066 | −0.050 | 0.825 |

$* p < 0.05; ** p < 0.01; *** p < 0.001.$

## 4. Discussion

This study aims to evaluate the clinical efficiency of maxillary and mandibular total arch distalization by analyzing and investigating the stability of TAD-assisted total arch distalization in adult patients. In an earlier study on mandibular molar distalization



using miniplates, it was reported that its stability was maintained for over a year [25]. However, no cases have yet been reported on the stability of total arch distalization using skeletal anchorage. To assess the stability of maxillary and mandibular total arch distalization, we used pretreatment (T0), post-treatment (T1), and postretention (T2) lateral cephalometric radiographs and dental casts. The use of dental casts along with lateral cephalometric radiographs enabled evaluation of both transverse dental changes and anteroposterior movements.

The subjects of this study were patients who were treated with total arch distalization using TADs, i.e., distalization of the anterior and posterior parts of dentition as one unit using TADs. T1 lateral cephalometric radiographs and dental casts revealed statistically significant distalization of both incisors and molars. The maxillary and mandibular first molars were distalized by 2.46 and 2.60 mm on average, respectively, which are lower results than those yielded in the study of Yamada et al. and similar to those reported by Oh et al., who conducted total arch distalization using the same method [26,27]. These interstudy differences may be ascribed to the different treatment goals depending on the severity of anterior crowding and the degree of required correction of the molar relationship. T2 lateral cephalometric radiographs and diagnostic casts revealed that all teeth underwent mesial drift during the retention period, whereby statistically significant mesial drift was observed in maxillary first molar crowns (mean 0.52 mm) and maxillary second molar crowns and roots (mean 0.65 and 0.58 mm, respectively). Sugawara et al. reported that the mean mesial drift of mandibular molars was 0.3 mm (range: 0.0–1.0 mm) one year after mandibular molar distalization using miniplates [25]. Akgul and Toygar performed a long-term observational study of tooth movements in adults without orthodontic treatment and reported mesial drift, although statistically not significant, of maxillary molars (0.42 mm in women and 0.26 mm in men on average) and incisors (0.07 mm in women, 0.39 mm in men) [28]. These results are largely consistent with the mesial drifting tendency observed in this study, suggesting that postdistalization mesial drifting tendency does not exceed the mesial drifting tendency in untreated adults.

One of the arguments advocating extraction treatment is that maxillary and mandibular molars cannot be distalized bodily, especially after the eruption of the second molars [29]. However, it was found that maxillary molar distalization using intraoral appliances can achieve first molar distal tipping of 5.4°, thus providing tooth movement close to bodily movement [30]. Distalization using skeletal anchorage was also reported to result in distalization close to bodily movement. Park et al. reported maxillary first and second molar distal tipping of 0.31° and 2.06°, respectively, and mandibular first and second molar distal tipping of 4.95° and 8.61°, respectively. Likewise, the results of this study revealed maxillary first and second molar distal tipping of 2.66° and 0.50°, respectively, and mandibular first and second molar distal tipping of 2.30° and 7.21°, respectively, similar to bodily movement [22]. Oh et al. noted that single tooth distalization is prone to untoward rotation or tipping when the force does not pass through the center of resistance, whereas application of distal force throughout the entire dentition can reduce such adverse effects because the teeth move under a rigid archwire engagement [26]. On the other hand, Fudalej and Antoszewska pointed out that molar distalization using miniscrew can induce more distal tipping than distalization using dental anchorage, and attributed it to the fact that a distalization appliance using dental anchorage exerts less force posteriorly due to mesial movement of anterior anchorage, whereas the stable anchorage provided by molar distalization using a miniscrew continuously transmits the total force to the posterior teeth [31].

The maxillary second molar underwent less distal tipping than the first molar, presumably because second molars are often distally impacted in the untreated state and undergo mesial tipping during teeth leveling. Of 28 patients, sixteen showed post-treatment mesial tipping of the second molars. The mandibular molars underwent more distal tipping than maxillary molars, especially the mandibular second molars. The anatomical structure of the mandible, which includes the lingual cortex, can be an obstacle for distalizing mandibular

molars, which causes distal tipping rather than bodily movement [32]. Ghosh and Nanda reported that distalization using a distal jet resulted in the distal tipping of the first and second molars (8.36° and 11.99°, respectively), noting that the molar relationship could be corrected but its retention stability was questioned [14]. Oh et al. predicted high-stability treatment for total arch distalization on the ground of the posterior movement close to bodily movement [26]. As shown in this example, the degree of tipping movement of the posterior region is generally believed to be correlated with the stability of treatment outcome. The results of this study also verify the correlation ($r = -0.634$, $p < 0.01$) between the degrees of distal tipping of the mandibular second molar during the treatment period and its mesial tipping during the retention period as a result of the correlation analysis of the changes in the molar angulation during the treatment and retention periods. In other words, the larger the distal tipping during the treatment period, the larger the mesial tipping during the retention period, which underlines the need to induce molar bodily movement to ensure a stable treatment outcome.

The upper and lower lips relative to the E-line moved distally after distal retraction of the anterior teeth by 0.89 and 1.06 mm, respectively. Although mesial drift of the maxillary and mandibular incisor was observed during the retention period, the upper and lower lips did not change significantly, which means that the distalization effect was stable from the viewpoint of the soft tissue profile. Bishara et al. reported a change in the soft tissue profile between the ages of 15 and 25, with the upper and lower lips relative to the E-line moving distally, and the same tendency between 25 and 45 years of age, so that such soft tissue change could affect the determination of the extraction and non-extraction [33]. Therefore, considering the fact that the lips move distally with increasing age, it is efficient to resolve crowding and improve the facial profile using the TADs without extraction, which will provide a better facial profile in the long term.

The results of careful observation of post-treatment changes in both anterior and posterior segments of distalization suggest that a method of distal movement of the entire dentition using TADs is advantageous over a method using intraoral distalization appliances such as a pendulum or distal jet, because skeletal anchorage prevents anterior anchorage loss. Even in case of anterior arch crowding, distalization of the buccal segment can provide alignment space for the anterior segment so that arch distalization can be performed after anterior alignment, thus preventing round tripping of the anterior segment. Oh et al. reported that total arch distalization can shorten the treatment period compared with distalization using an intraoral appliance despite slower movement of each individual tooth, because all teeth are simultaneously distalized [26]. The mean treatment period in this study was 24.5 ± 9.6 months, similar to 20.0 ± 4.9 months reported by Oh et al.

In Yamada et al., who performed maxillary arch distalization using the similar method, posterior intrusion and anterior extrusion were observed [27]. In our study, however, both anterior and posterior sections showed intrusion. The mean amount of intrusion of the upper central incisors was 0.28 mm, which was not statistically significant, and those of the first and second molars were 0.92 and 0.89 mm, respectively, which were statistically significant. This interstudy difference is presumably due to the fact that our study included cases using two miniscrews bilaterally, whereas only one miniscrew was used between the second premolar and first molar bilaterally in the study of Yamada et al. A finite element analysis study reported that the center of resistance of the maxillary dentition is located between the roots of the upper premolars [34] and that intrusion of the entire maxillary dentition can be achieved by applying force to it with two miniscrews bilaterally [35]. This intrusion of the posterior segment is considered to have contributed to maintaining the post-treatment stability without incurring an increase in mandibular angle.

During the retention period, all teeth showed a statistically non-significant extrusion tendency (incisor: 0.26 mm, first molar: 0.35 mm, second molar: 0.38 mm). Akgul and Toygar reported a similar or more marked extrusion tendency in adult males without orthodontic treatment in a long-term observational study (incisor: 0.33 mm, molar: 0.63 mm) [28]. Baek et al. examined the stability of anterior open-bite correction with intru-

sion of maxillary posterior teeth using miniscrews and reported a relapse rate of 22.88% (0.45 mm on average) for the intruded maxillary first molars [36]. As demonstrated by these reports, a certain degree of extrusion can occur during the retention, and additional orthodontic treatment may be necessary to maintain the vertical position of the posterior segment in cases where the vertical change in the posterior segment is determinant for maintaining the stability of treatment outcome.

The values of PTV-B decreased slightly before treatment, after treatment, and during the post-treatment period. The posterior movement after treatment at point B was 0.35 mm, which was statistically significant, suggesting that the anterior alveolar bone was absorbed and the alveolar bone modeling occurred after the retraction of anterior teeth.

Analysis of dental casts revealed an increase in the arch width during the treatment period, whereby the intercanine and first molar widths increased with statistical significance. Figures 9 and 10 gives superimpositions of the digital images of the initial and final casts. Oh et al. reported a statistically significant increase in the width of the posterior segment of the distalized dental arch [26]. They attributed the interpremolar width increase to the force vector acting in the buccal direction by the distal force applied on the premolar area through the TADs placed in the buccal posterior region and the intermolar width increase to the buccal tipping of the molars by the intrusion force applied on the buccal segment bracket. This can be prevented by using a rigid archwire with a slight constriction around the canines or passive trans palatal arch. The basal bone arch, which becomes wider posteriorly, is also considered to contribute to arch width increase during total arch distalization.

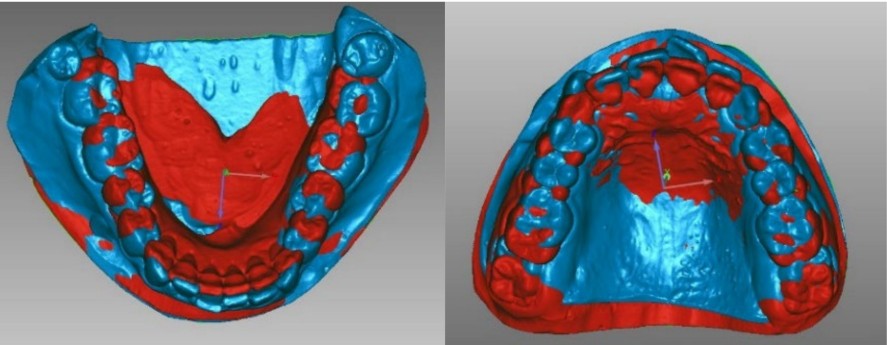

**Figure 9.** Maxillary and mandibular superimpositions of pretreatment (blue) and post-treatment (red) digital dental models of a patient showing expansion of the dental arch.

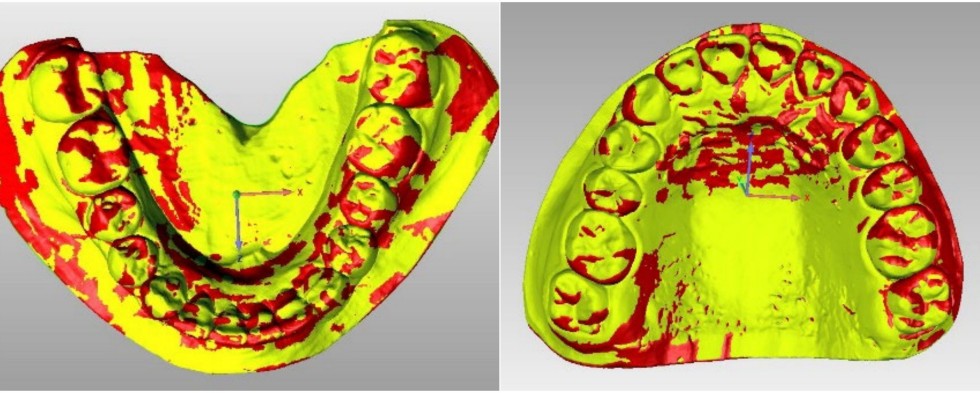

**Figure 10.** Maxillary and mandibular superimpositions of post-treatment (red) and postretention (yellow) digital dental models of a patient showing stable retention.

During the retention period, statistically non-significant decreases in the arch width were shown at all measurement points. Park et al. reported that arch width was maintained stably 16 years after orthodontic treatment with or without extraction [37]. In contrast,

Miyazaki et al. reported a mean increase of 0.99 mm in the maxillary intercanine arch width after extraction treatment in adult patients and a mean decrease of 0.39 mm during the retention period [38]. The result of this study shows that the increase in the width of the arches was better maintained than that of the other treatments using TADs. This may be due to the fact that the upper and lower dentitions during the treatment period are occluded and simultaneously moved backwards to help stabilize the width of the arches.

Vertical skeletal parameters have been considered a factor affecting the stability of treatment outcome [39]. However, Zaher et al. reported that vertical skeletal parameters have no significant influence on the stability of orthodontic treatment outcome [40]. Our correlation analysis revealed no statistically significant correlation between the mandibular plane angle and facial height ratio at the initial status and the teeth movements during the retention period. However, a negative correlation was found between only the ANB angle at the initial status and the amount of maxillary incisor. First and second molar crown mesial drift during the retention period can possibly be explained by the fact that the anteroposterior skeletal factor is more correlated with relapse than the vertical factor. In this regard, further study and investigation of relapse-related parameters are considered necessary.

As regards relapse rate, mean horizontal relapse rates of 16.0%, 21.1%, and 28.8% were exhibited by the incisor crown, first and second molar crowns, respectively. What is worth noting here is the high interpatient variability; for example, the first molar relapse rates ranged between 0.79 and 43.84%, and the amounts of mesial drift between 0.02 and 0.87 mm. Further research is necessary to find out the causes of this high interpatient variability and to identify related indicators, which will greatly contribute to developing personalized orthodontic options to reduce overcorrection and relapse.

Previous studies on the stability of orthodontic treatment were primarily carried out with pediatric patients, relying on dental casts and the peer assessment rating index or Little's irregularity index for stability evaluation. Therefore, we could not find any study to directly check the results of our study against. The dental casts and index alone are barely enough to determine the relative movements of maxillary and mandibular teeth, and cannot be used to identify the place of relapse. On this note, the significance of this study lies in the fact that it used lateral cephalometric radiographs along with diagnostic casts for the evaluation of the stability of orthodontic treatment in adult patients.

The limitations of this study are small sample size, measurements made at the midpoints of the opposite-side teeth, and image overlap with the maxilla and mandible due to the use of radiographs taken in a closed mouth position. Sugawara et al. used lateral cephalometric radiographs taken in an open mouth position for easier and clearer identification of individual teeth [41]. Additionally, this study included buccal, midpalatal and retromolar miniscrews, which required different biomechanics to obtain the wanted distalization. Prospective studies with a larger number of subjects need to be carried out as follow-up studies.

## 5. Conclusions

The purpose of this study was to quantify the treatment effects and post-treatment stability of total arch distalization with TADs in adults, and the outcomes were as follows:

1.  The maxillary incisor ($2.68 \pm 2.19$ mm), first molar ($2.46 \pm 1.97$ mm) and second molar ($2.60 \pm 2.52$ mm) were significantly distalized after treatment ($p < 0.001$). Intrusion of the maxillary first ($0.92 \pm 1.16$ mm) and second molars ($0.89 \pm 1.17$ mm) was also observed after the treatment ($p < 0.01$), which presumably caused a decrease in the distance from ANS to Me. The mandibular first molar ($2.57 \pm 2.13$ mm, $p < 0.01$) and second molar ($2.24 \pm 2.35$ mm, $p < 0.05$) were significantly distalized after treatment.

2.  During the retention period, significant mesial movement of the maxillary first molar ($0.52 \pm 0.99$ mm) and second molar ($0.65 \pm 0.92$ mm) was observed ($p < 0.05$); however, intrusion was kept relatively stable. Mesial movement of the mandibular arch was also observed but was not statistically significant during the retention period.

3. There were no changes in skeletal measurements after distalization except the decrease in distance from ANS to Me and PTV to B.

4. The upper and lower lip were retracted by $0.89 \pm 1.19$ mm ($p < 0.001$) and $1.06 \pm 1.91$ mm ($p < 0.05$), respectively, and there was no significant relapse during the retention period.

5. The maxillary intercanine and intermolar width increased by $1.52 \pm 1.63$ mm ($p < 0.001$) and $0.93 \pm 1.21$ mm ($p < 0.01$), respectively, on average, after the treatment. The arch width was relatively stable without significant changes during the retention period. Distal-in rotation of the molars was observed after the treatment, and there were no significant changes during the retention period

6. Post-treatment changes of distalized teeth were correlated with the amount of distalization during treatment but not with the initial skeletal pattern and retention period.

It was concluded that even though there was a little relapse in the anteroposterior position of the maxillary and mandibular teeth during retention, there was no obvious relapse in facial profile. Therefore, the total arch distalization can be used in patients with a moderate amount of arch length discrepancy effectively with stable retention.

**Author Contributions:** Conceptualization, B.-J.S. and H.-S.Y.; methodology, H.-S.Y.; software, B.-J.S.; validation, K.-J.L., J.-Y.C., J.-S.L. and S.-S.M.; formal analysis, H.-S.Y.; investigation, B.-J.S.; resources, H.-S.Y.; data curation, B.-J.S.; writing—original draft preparation, B.-J.S.; writing—review and editing, H.-S.Y.; visualization, B.-J.S.; supervision, H.-S.Y.; project administration, H.-S.Y. All authors have read and agreed to the published version of the manuscript.

**Funding:** This research received no external funding.

**Institutional Review Board Statement:** The Yonsei Dental Hospital institutional review board (CRNo: 2-2020-0013) approved this study.

**Informed Consent Statement:** Informed consent was obtained from all subjects involved in the study.

**Data Availability Statement:** The data presented in this study are available on request from the corresponding author. The data are not publicly available due to privacy issue.

**Conflicts of Interest:** The authors declare no conflict of interest.

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
