# Peer review of "Stability of the Maxillary and Mandibular Total Arch Distalization Using Temporary Anchorage Devices (TADs) in Adults"

_applsci, doi:10.3390/app12062898_

Round 1

Reviewer 1 Report

This article has a very current and interesting topic for the scientific community.
It is well structured and easy to read. The introduction, discussion and conclusions are well written and with all relevant points highlighted.
Authors should only make small corrections in materials and methods and in the results:
1. Authors must present and refer in the text to the study approval document by the ethics committee.
2. Authors must present the potency test performed for the chosen sample number.
3. Authors should transform tables 1, 3, 4 and 5 into graphs for better reading and interpretation

Author Response

1. Authors must present and refer in the text to the study approval document by the ethics committee.

 -> The Yonsei Dental Hospital institutional review board (CRNo: 2-2020-0013) approved this study, and informed consent agreements were signed by the participants.

2. Authors must present the potency test performed for the chosen sample number.

  ->The minimum sample size was calculated using G*Power 3 (Düsseldorf, Germany) with a significance level of p<0.05 and a power of 80%, and confirmed as 25. 

3. Authors should transform tables 1, 3, 4 and 5 into graphs for better reading and interpretation 

  -> I think that is a good point. At first, I also thought that the values measured in this study were too many to be viewed in a table. So I had a lot of trouble figuring out how to solve this.

However, there were some points that could only be expressed in tables.

     As can be seen from the table, Table 1 is thought to be difficult to convert into a graph because it contains the characteristics of each patient, such as age and location of TADs placement.

     Tables 3, 4, and 5 have too many measurement parameters, so it is rather inconvenient to look at them as graphs.  However, if it is better to express it with a graph and figure, I think it would be better to display only the dental changes in the molars and anterior teeth, since total distalization mainly results in dental changes rather than skeletal changes. Otherwise, it seems that there are too many figures and graphs. 

     I think it is easier to see the results using the table if you only look at the numbers with significant results in the table.

Reviewer 2 Report

The paper itself is well written and documented, showing a great effort from the authors.

The topic sounds original and with an interesting clinical meaning.

I also would make only the following few mentions:

  • Did you take any informed consent from the patients? Did you have an ethics committee approval?
  • Line 86: which type of TADs did you use? Please specify.
  • Lines 95-102: in this paragraph you explained where the miniscrews were positioned. The study included both buccal miniscrews, midpalatal miniscrews and retromolar miniscrews which required different biomechanics to obtain the wanted distalization. Each biomechanics should be separately investigated comparing the buccal systematic and the palatal systematic in the maxillary arch and the buccal systematic and the retromolar systematic in the mandibular arch. Your sample included more buccaly positioned anchorage so I suggest to investigate only this patients eliminating the subjects with palatal or retromolar anchorage.
  • I suggest to add some picture of the clinical casas with miniscrews positioned.
  • Line 203: you reported the model measurement without differentiating the group who underwent mandibular or maxillary arch distalization as you did in the cephalometric parameters. Please separate the two groups.
  • Did you have any failure of the miniscews? Please specify.

Author Response

·         Did you take any informed consent from the patients? Did you have an ethics committee approval?

     ->  The Yonsei Dental Hospital institutional review board (CRNo: 2-2020-0013) approved this study, and informed consent agreements were signed by the participants.

·         Line 86: which type of TADs did you use? Please specify.

     ->  miniscrews (Ortholution, Seoul, Korea) were used.

·         Lines 95-102: in this paragraph you explained where the miniscrews were positioned. The study included both buccal miniscrews, midpalatal miniscrews and retromolar miniscrews which required different biomechanics to obtain the wanted distalization. Each biomechanics should be separately investigated comparing the buccal systematic and the palatal systematic in the maxillary arch and the buccal systematic and the retromolar systematic in the mandibular arch. Your sample included more buccaly positioned anchorage so I suggest to investigate only this patients eliminating the subjects with palatal or retromolar anchorage.

     -> Thanks for the good point. I was also aware of that part and thought a lot about how to solve it.

However, since the number of patients is small, there may be slight fluctuations in the placement position in terms of treatment effectiveness, but what we mainly focus on is retention, and the placement position does not have much to do with retention. 

        Also, unlike the general perception, as can be seen from the study results, the placement position is not significant in terms of treatment results and especially retention. 

        In addition, because the reply period was short, I thought that time was running out to process statistics again.

        However, since it is clear that there is a biomechanical difference in the placement position, this will be described as one of the limitations of this study in the discussion section.

·         I suggest to add some picture of the clinical casas with miniscrews positioned.

   -> I will include some pictures of partial canine retraction and en masse retraction using miniscrews.

·         Line 203: you reported the model measurement without differentiating the group who underwent mandibular or maxillary arch distalization as you did in the cephalometric parameters. Please separate the two groups.

  -> Referring to Table 5, the group were differentiated who underwent maxillary or mandibular arch distalization as maxilla (Mx) and mandible (Mn), as indicated by the cephalometric parameters.

·         Did you have any failure of the miniscews? Please specify.

  -> In very few cases, screw placement failed. In case the screw fails, re-implantation proceeds as soon as possible so that the entire treatment period is not affected.

Round 2

Reviewer 2 Report

The authors have sufficiently improved the article and gave an exhaustive answer to my comments.

My main comment about the protocol has been added in the limitations of the study.

I think the paper is now suitable for pubblication.